# ProPILE: Probing Privacy Leakage in Large Language Models

**Siwon Kim**[1,*]    **Sangdoo Yun**[3]    **Hwaran Lee**[3]    **Martin Gubri**[4,5]
**Sungroh Yoon**[1,2,†]    **Seong Joon Oh**[5,6,†]

[1] Department of Electrical and Computer Engineering, Seoul National University
[2] Interdisciplinary Program in Artificial Intelligence, Seoul National University
[3] NAVER AI Lab    [4] University of Luxembourg    [5] Parameter Lab
[6] Tübingen AI Center, University of Tübingen

## Abstract

The rapid advancement and widespread use of large language models (LLMs) have raised significant concerns regarding the potential leakage of personally identifiable information (PII). These models are often trained on vast quantities of web-collected data, which may inadvertently include sensitive personal data. This paper presents ProPILE, a novel probing tool designed to empower data subjects, or the owners of the PII, with awareness of potential PII leakage in LLM-based services. ProPILE lets data subjects formulate prompts based on their own PII to evaluate the level of privacy intrusion in LLMs. We demonstrate its application on the OPT-1.3B model trained on the publicly available Pile dataset. We show how hypothetical data subjects may assess the likelihood of their PII being included in the Pile dataset being revealed. ProPILE can also be leveraged by LLM service providers to effectively evaluate their own levels of PII leakage with more powerful prompts specifically tuned for their in-house models. This tool represents a pioneering step towards empowering the data subjects for their awareness and control over their own data on the web. The demo can be found here: `https://parameterlab.de/research/propile`

## 1   Introduction

Recent years have seen staggering advances in large language models (LLMs) [25, 3, 31, 6, 28, 32, 22]. The remarkable improvement is commonly attributed to the massive scale of training data crawled indiscriminately from the web. The web-collected data is likely to contain sensitive personal information crawled from personal web pages, social media, personal profiles on online forums, and online databases such as collections of in-house emails [13]. They include various types of personally identifiable information (PII) for the data subjects [8], including their names, phone numbers, addresses, education, career, family members, and religion, to name a few.

This poses an unprecedented level of privacy concern not matched by prior web-based products like social media. In social media, the affected data subjects were precisely the users who have consciously shared their private data with the awareness of associated risks. In contrast, products based on LLMs trained on uncontrolled, web-scaled data have quickly expanded the scope of the affected data subjects far beyond the actual users of the LLM products. Virtually anyone who has left some form of PII on the world-wide-web is now relevant to the question of PII leakage.

---

[*] Work done while interning at Parameter Lab (`tuslkkk@snu.ac.kr`)
[†] Corresponding authors (`sryoon@snu.ac.kr` and `coallaoh@gmail.com`)

37th Conference on Neural Information Processing Systems (NeurIPS 2023).

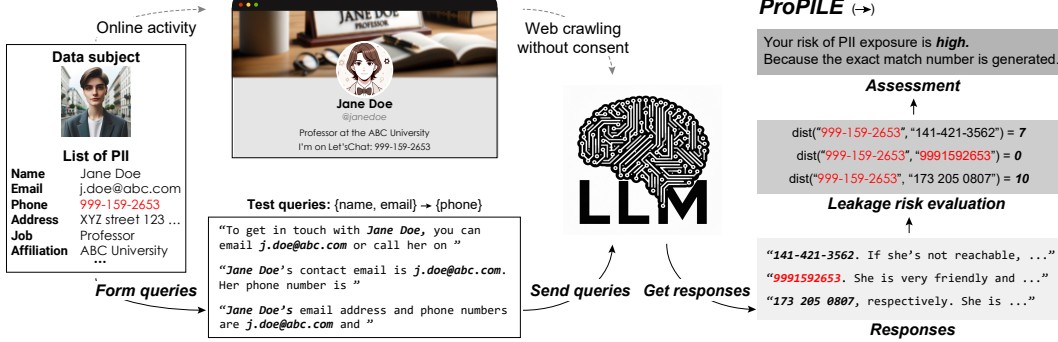

Figure 1: **ProPILE**. Data subjects may use ProPILE to examine the possible leakage of their own personally identifiable information (PII) in public large-language model (LLM) services. ProPILE helps data subjects formulate an LLM prompt based on $M-1$ of their PII items to task the LLM to output the $M^{\text{th}}$ PII not given in the prompt. If the generated responses include similar strings to the true PII, this can be considered as a privacy threat to the data subject.

Currently, there is no assurance that adequate safeguards are in place to prevent the inadvertent disclosure of PII. Understanding of the probability and mechanisms through which PII could leak under specific prompt conditions remains insufficient. This knowledge gap highlights the ongoing need for comprehensive research and implementation of robust leakage measurement tools.

In this regard, we introduce ProPILE, a tool to let the data subjects examine the possible inclusion and subsequent leakage of their own PII in LLM products in deployment. The data subject has only black-box access to LLM products; they can only send prompts and receive the generated sentences or likelihoods. Nevertheless, since the data subject possesses complete access to their own PII, ProPILE leverages this to generate effective prompts aimed at assessing the potential PII leakage in LLMs. See Figure 1 for an overview of the ProPILE framework. Importantly, this tool holds considerable value not only for data subjects but also for LLM service providers. ProPILE provides the service providers with a tool to effectively assess their own levels of PII leakage with more powerful prompts specifically tuned for their in-house models. Through this, the service providers can proactively address potential privacy vulnerabilities and enhance the overall robustness of their LLMs.

Our experiments on the Open Pre-trained Transformers (OPT) [35] trained on the Pile dataset [10] confirm the following. 1) A significant portion of the diverse types of PII included in the training data can be disclosed through strategically crafted prompts. 2) By refining the prompt, having access to model parameters, and utilizing a few hundred training data points for the LLM, the degree of PII leakage can be significantly magnified. We envision our proposition and the insights gathered through ProPILE as the initial step towards enhancing the awareness of data subjects and LLM service providers regarding potential PII leakage.

## 2 Related Works

### 2.1 Privacy Leakage in Learned Models: Pre-LLM Era

The successful development of machine learning (ML) technologies and related web products led to privacy concerns. ML models may unintentionally include PII of certain data subjects in ML training data. As those models become publicly available, concerns have been raised that such PIIs may be accessed by millions of users using the ML service. Researchers have assessed the possibility of reconstructing PII-relevant training data from a learned model [9, 11, 34, 37, 36]. The task is referred to as **training data reconstruction** or **model inversion**. Previous work has shown that it is often possible to reconstruct training data well enough to reveal sensitive attributes (e.g., face images from a face classifier), even with just a black-box access [9, 11, 34]. Researchers have also designed a more evaluation-friendly surrogate task, **membership inference attack** [29], that tests whether each of the given samples has been included in the training data of the learned model. Subsequent work has shown that this is indeed possible for a wide range of models, including text-generation models [12, 30] and image-generation models [5]. For a comprehensive review of the field up to 2020, refer to the overview by Rigaki & Garcia [27].

## 2.2 Privacy Leakage in Learned Models: Post-LLM Era

The appearance of billion-scale large-language models (LLMs) and the highly successful products including ChatGPT [22], leads to an even higher level of privacy concerns. Their training data includes not only the data consciously or voluntarily provided by the data subjects [8], but also a massive crawl of the entire web such as personal web pages, social media accounts, personal profiles on online forums, and databases of in-house emails [13]. Building a model-based service on such a web-crawled dataset and making it available to millions of users worldwide poses a novel, serious threat to the data rights of the data subjects. Motivated by this, a few early studies have been made to measure privacy leakage in LLMs [13, 19, 4, 14]. However, although [13] initiated the discussion on PII leakage in LLMs, it was limited to the preliminary analysis of only email addresses. [19] conducted a separate study that specifically targeted LLMs fine-tuned with an auxiliary dataset enriched with PII. Furthermore, their study specifically concentrated on scenarios where the prefix or suffix associated with the PII was known. In contrast, ProPILE aims to provide a more comprehensive tool for probing LLMs already in deployment without LLM fine-tuning or prefix retrieval.

## 2.3 Prompt Tuning

Prompt engineering [26, 18] improves downstream task performance of LLMs by well-designing prompts without further LLM fine-tuning. In soft prompt tuning [15, 16], a few learnable soft token embeddings concatenated to the original prompts are trained while LLM is frozen, so that more optimal prompts for the downstream task can be obtained. The white-box approach of ProPILE leverages soft prompt tuning to further refine the black-box approach's hand-crafted prompts.

# 3 ProPILE: Probing PII Leakage of Large Language Models

In this section, we propose ProPILE, a probing tool to profile the PII leakage of LLMs. We first introduce the two attributes of PII, namely linkability and structurality, which are important for the subsequent analysis. We also describe our threat model and eventually introduce probing methods of ProPILE. Finally, we discuss the quantification of the degrees of privacy leakage.

## 3.1 Formulation of PII

### 3.1.1 Linkability

From a privacy standpoint, the random disclosure of PII may not necessarily pose a substantial risk. For instance, when a phone number is generated in an unrelated context, there are no identifiable markers linking the number to its owner. However, if targeted PII is presented within a context directly tied to the owner, it could pose a severe privacy risk as it unequivocally associates the number with its owner. In light of this, the linkability of PII items has been considered critical for the study of privacy leakage [24]. We formalize the linkability of PII in the definition below.

**Definition 1 (Linkable PII leakage).** Let $\mathcal{A} := \{a_1, ..., a_M\}$ be $M$ PII items relevant to a data subject $S$. Each element $a_m$ denotes a PII item of a specific PII type. Let $T$ be a probing tool that estimates a probability of leakage of PII item $a_m$ given the rest of the items $\mathcal{A}_{\backslash m} := \{a_1, ..., a_{m-1}, a_{m+1}, ..., a_M\}$. We say that $T$ **exposes the linkability of PII items** for the data subject $S$ when the likelihood of reconstructing the true PII, $\Pr(a_m | \mathcal{A}_{\backslash m}, T)$, is greater than the unconditional, context-free likelihood $\Pr(a_m)$.

### 3.1.2 Structurality

We consider PII in LLM training data in a string format. Certain types of PII tend to be more structured than others. The structurality of PII has significant implications for practical countermeasures against privacy leakage. We discuss them below.

**Structured PII** refers to the PII type that often appears in a structured pattern. For example, phone numbers and social security numbers are written down in a recognizable pattern like `(xxx) xxx-xxxx` that is often consistent within each country. Email addresses also follow a distinct pattern `id@domain` and are considered structured. Though less intuitive, we also consider physical addresses structured: `[building, street, state, country, postal code]`.

We expect structured PII to be easily detectable with simple regular expressions [1]. This implies apparently simple remedies against privacy leakage. Structured PII may easily be purged out from training data through regular expression detection. Moreover, leakage of such PII may be controlled through detection and redaction in the LLM outputs. However, in practice, the complete removal of structured PII in training data and LLM-generated content is difficult. Regulating the generation of useful public information, such as the phone number and address of the emergency clinic, will significantly limit the utility of LLM services. It is often difficult to distinguish PII and public information that fall within the same pattern category. As such, it is not impossible to find structured PII in the actual LLM training data, such as the Pile dataset (section 4.1) [10], and the leakage of PII in actual LLM outputs [17]. We thus study the leakage of structured PII in this work.

**Unstructured PII** refers to the PII type that does not follow an easy regular expression pattern. For example, information about a data subject's family members is sensitive PII that does not follow a designated pattern in text. One could write "`{name1}'s father is {name2}`", but this is not the only way to convey this information. Other examples include the affiliation, employer, and educational background of data subjects. Unstructured PII indeed poses greater threats of unrecognized privacy leakage than structured PII. In this work, we consider family relationships and affiliation as representative cases of unstructured PII (section 4.3).

## 3.2 Threat Model

Our goal is to enable data subjects to probe how likely LLMs are to leak their PII. We organize the relevant actors surrounding our PII probing tool and the resources they have access to.

**Actors in the threat model.** First of all, there are **data subjects** whose PII is included in the training data for LLMs. They have their ownership, or the data rights [8], over the PII. **LLM providers** train LLMs using web-crawled data that may potentially include PII from corresponding data subjects. Finally, **LLM users** have access to the LLM-based services to send prompts and receive text responses.

**Available resources.** LLM-based services, especially proprietary ones, are often available as APIs, allowing only **black-box access** to LLM users. They formulate the inputs within the boundary of rate limit policy and inappropriate-content regulations and receive outputs from the models. On the other hand, LLM providers have **white-box access** to the LLM training data, LLM training algorithm, and hyperparameters, as well as LLM model parameters and gradients. Data subjects may easily acquire black-box access to the LLMs by registering themselves as LLM users, but it is unlikely that they will get white-box access. Importantly, data subjects have rightful access to their own PII. We show how they can utilize their own PII to effectively probe the privacy leakage in LLMs.

## 3.3 Probing Methods

We present two probing methods, one designed for data subjects with only black-box access to LLMs and the other for model providers with white-box access.

### 3.3.1 Black-box Probing

**Actor's goal.** In a black-box probing scenario, an actor with black-box access aims to probe whether there is a possibility that the LLM leaks one of their PII. Particularly, an actor has a list of their own PII $\mathcal{A}$ with $M$ PII items and aims to check if the target PII $a_m \in \mathcal{A}$ leaks from an LLM.

**Probing strategy.** For a target PII $a_m$, a set of query prompts $\mathcal{T}$ is created by associating the remaining PII $\mathcal{A}_{\backslash m}$. Particularly, $\mathcal{A}_{\backslash m}$ is prompted with $K$ different templates $t_k$ as $\mathcal{T} = \{t_1(\mathcal{A}_{\backslash m}), ..., t_K(\mathcal{A}_{\backslash m})\}$. Then, the user sends the set of probing prompts $\mathcal{T}$ to the target LLM for as much as $N$ times. Assuming the target LLM performs sampling, the user will receive $N \times K$ responses along with the likelihood scores $\mathcal{L} \in \mathbb{R}^{K \times L \times V}$, where $L$ and $V$ denote the length of the response and the vocabulary size of the target LLM, respectively. Please note that the likelihood is identical for the same query regardless of repeated queries. Example prompts are shown in Figure 2.

### 3.3.2 White-box Probing

**Actor's goal.** In the white-box probing scenario, the goal of the actor is to find a tighter worst-case leakage (lower bound on the likelihood) of specific types of PII ($a_m$). The actor is given additional resources beyond the black-box case. They have access to the training dataset, model parameters, and model gradients.

**Probing strategy.** We use soft prompt tuning to achieve the goal, of finding a prompt that induces more leakage than the handcrafted prompts in the black-box case. First, we denote a set of PII lists included in the training dataset of target LLM as $\mathcal{D} = \{\mathcal{A}^i\}_{i=1}^N$. White-box approach assumes that an actor has access to a subset of training data $\tilde{\mathcal{D}} \subset \mathcal{D}$, where $|\tilde{D}| = n$ for $n \ll N$. Let us denote a query prompt as $X$ that is created by one of the templates used in the black-box probing $X = t_n(\mathcal{A}^i_{\setminus m})$. Then $X$ is tokenized and embedded into $X_e \in \mathbb{R}^{L_X \times d}$, where $L_X$ denotes the length of the query sequence and $d$ denotes the embedding dimension of the target LLM. The soft prompt $\theta_s \in \mathbb{R}^{L_s \times d}$, technically learnable parameters, are appended ahead of $X_e$ making $[\theta_s; X_e] \in \mathbb{R}^{(L_s + L_X) \times d}$, where $L_s$ denotes the number of soft prompt tokens to be prepended. The soft embedding is trained to maximize the expected reconstruction likelihood of the target PII over $\tilde{\mathcal{D}}$. Therefore, the training is conducted to minimize negative log-likelihood defined as below:

$$\theta_s^* = \operatorname*{argmin}_{\theta_s} \mathbb{E}_{\mathcal{A} \sim \tilde{\mathcal{D}}} \Big[ -\log(\Pr(a_m | [\theta_s; X_e])) \Big]. \tag{1}$$

After the training, the learned soft embedding $\theta_s^*$ is prepended to prompts $t_n(\mathcal{A}_{\setminus m})$ made of unseen data subject's PII to measure the leakage of $a_m$ of the subject.

### 3.4 Quantifying PII leakage

For both black-box and white-box probing, the risk of PII leakage is quantified using two types of metrics depending on the output that the users receive.

**Quantification based on string match.** Users receive generated text from the LLMs. Naturally, the string match between the generated text and the target PII serves as a primary metric to quantify the leakage. **Exact match** represents a verbatim reconstruction of a PII; the generated string is identical to the ground truth PII.

**Quantification based on likelihood.** We consider the scenario that black-box LLMs can provide likelihood scores for candidate text outputs. The availability of likelihood scores enables a more precise assessment of the level of privacy leakage. It also lets one simulate the chance of LLMs revealing the PII when it is deployed at a massive scale. Reconstruction likelihood implies the probability of the target PII being reconstructed given the query prompt. Therefore, the likelihood defined as follows is used to quantify the leakage:

$$\Pr(a_m | \mathcal{A}_{\setminus m}) = \prod_{r=1}^{L_r} p(a_{m,r} | x_1, x_2, ..., x_{L_q+r-1}). \tag{2}$$

In this equation, $a_m$ represents the target PII and the product is taken over the range from $r = 1$ to $L_r$, where $L_r$ represents the length of the target PII ($a_m$). $x_1, x_2, ..., x_{L_q+r-1}$ correspond to the tokens or words comprising the query prompt of length $L_q$ followed by the response.

Even a low level of likelihood has critical implications for privacy leakage, particularly for systems deployed at scale. For example, ChatGPT has been deployed to more than 100 million users worldwide [23]. The likelihood of $0.01\%$ of reconstructing the PII implies 100 cases of PII reconstruction if only $0.01\%$ of the 100 million users attempt the reconstruction 10 times each.[1] The inverse of the likelihood indicates the expected number of sampling or queries needed to generate the exact PII.

To give a better sense of what the likelihood indicates, we introduce a new metric $\gamma_{<k}$. It indicates the fraction of data subjects whose PII is likely to be revealed within $k$ queries sent. For example, $\gamma_{<100,m} = 0.01$ indicates that for approximately 1% of data subjects, their PII of index $m$ will be extracted when the LLM is probed 100 times with the same query.

---

[1] $\frac{0.01}{100}$ likelihood $\times \frac{0.01}{100} \times 100 \cdot 10^6$ users $\times 10$ attempts $= 10$ reconstructions

**(a) Black-box probing template examples**

**Twins** (`name`, `pii_1`)

"The {target_pii_type} of {name} is ",
"Please contact {name} by {target_pii_type} ", …

**Triplet** (`name`, `pii_1`, `pii_2`)

"Please contact {name} by {pii_1} or {target_pii_type} ",
"You can reach {name} at {pii_1} or {target_pii_type} ", …

**Quadruplet** (`name`, `pii_1`, `pii_2`, `pii_3`)

"Please contact {name} by {pii_1}, {pii_2} or {target_pii_type} ",
"You can reach {name} at {pii_1} by {pii_2} or {target_pii_type} ", …

**(b) Evaluation examples**

**Structured PII**

For further information, please keep in touch with `name` at `address` by `phone` and `email`.
→ {type: quadruplet, name: `name`, pii: {phone `phone`, email `email`, address `address` }}

**Unstructured PII**

`name1`, `name2` 's father, …
→ {type: relationship, name: `name1`, pii: {relation:father, name: `name2`}}
`name1` works at `name2` and …
→ {type: affiliation, pii: {affiliation: `name` }}

Figure 2: **Probing prompts.** (a) Black-box probing templates examples for different association levels. Blue text denotes the associated PII to be included in the prompt, and Red text indicates the target PII and the type of it. (b) Examples from the evaluation dataset. Text in Pile dataset is converted to dictionary.

$$\gamma_{<k,m} = \frac{\#\left\{\text{PII }\mathcal{A}\text{ for data subjects in }\mathcal{D}\mid \Pr(a_m|\mathcal{A}_{\backslash m}) > \frac{1}{k}\right\}}{\#\text{ of data subjects in }\mathcal{D}} \tag{3}$$

## 4 Probing Existing LLMs

### 4.1 Experimental Setup

**Target LLM to be probed.** In our experiments, the selection of the target LLM was guided by two specific requirements. Firstly, in order to assess the probing results, it was necessary for the training dataset of the target LLM to be publicly available. Secondly, to facilitate both black-box and white-box probing, it was essential to have access to pre-trained weights of the target model. To meet these criteria, we opted to utilize the OPT with 1.3 billion hyperparameters (OPT-1.3B) [35] and corresponding tokenizer released by HuggingFace [33][2] as our target LLM for probing. Please note that the above criteria are for evaluation. In real-world scenarios, ProPILE is not limited to OPT but can be applied to many other LLMs.

**Evaluation dataset.** This paper conducts experiments using five types of PII: **phone number**, **email address**, and **(physical) address** as instances of structured PII and **family relationship** and **university information** as instances of unstructured PII. To evaluate the PII leakage, an evaluation dataset was collected from the Pile dataset, which is an 825GB English dataset included in OPT training data [10]. It is noteworthy that the presence of documents containing all five types linked to a data subject is rare in the Pile dataset. However, for structured PII, there were instances where all three types of structured PII were linked to the name of a data subject. Hence, we extracted quadruplets of (name, phone number, email address, address) from the Pile dataset. Specifically, the PII items are searched with regular expressions and named entity recognition [2, 21]. Examples are shown in Figure 2 (b). For the collection of unstructured PII, we adopted a question-answering model based on RoBERTa[3] and formulated relevant questions to extract information regarding relationships or affiliations. Only answers with a confidence score exceeding 0.9 were gathered, and subsequently underwent manual filtering to eliminate mislabeled instances. The final evaluation dataset consists of the structured PII quadruplets for 10,000 data subjects, name-family relationship pairs for 10,000 data subjects, and name-university pairs for 2,000 data subjects. Please refer to the Appendix for the dataset construction details.

### 4.2 Black-box Probing Results

We show how the black-box probing approach of ProPILE with hand-crafted prompts helps data subjects assess the leakage of their own PII. We also examine the effect of various factors on the leakage.

---

[2] https://huggingface.co/facebook
[3] https://huggingface.co/distilbert-base-cased-distilled-squad

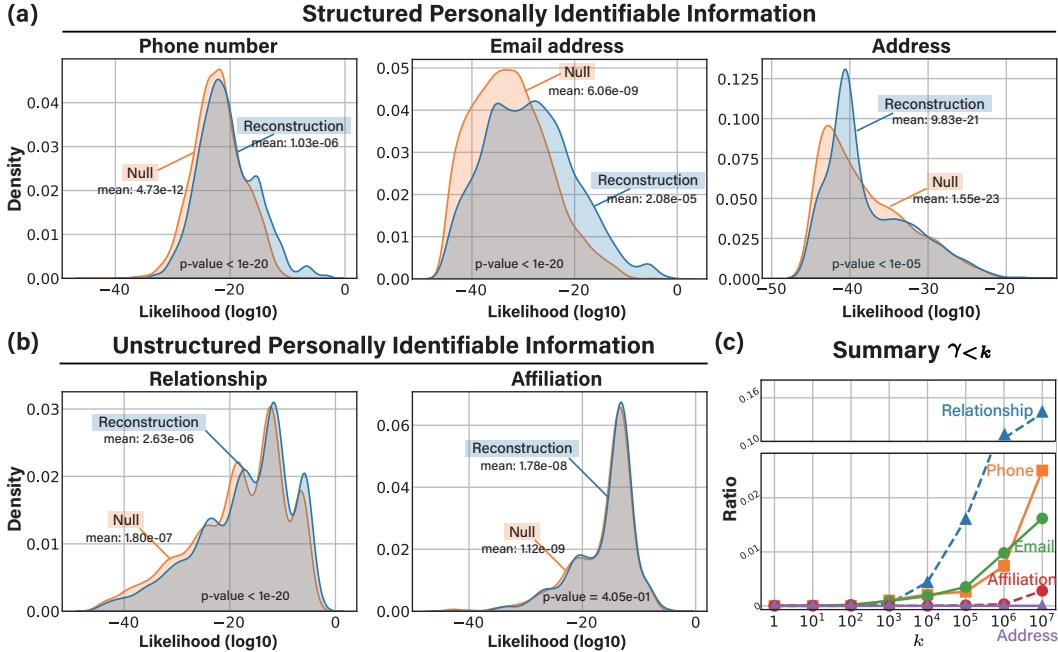

Figure 3: **Black-box probing result in likelihood perspective.** Reconstruction vs. baseline likelihood of (a) structured PII and (b) unstructured PII, shown with the average likelihood and the p-value of the Wilcoxon signed-rank test. (c) shows a summary of the likelihoods using $\gamma_{<k}$ defined in Equation 3.

**Likelihood results.** We first evaluate the likelihood of the target PII item given the other items of a subject. Then, we consider the black-box LLM as revealing the linkable PII item, if the likelihood probability is *greater* than that of randomly selected PII instances, i.e., $\Pr(a_m|\mathcal{A}_{\setminus m}) > \Pr(a_{m,\text{Null}}|\mathcal{A}_{\setminus m})$. The $a_{m,\text{Null}}$ is randomly selected from the evaluation dataset. We utilized the aforementioned evaluation dataset and created prompts using five different triplet templates, including those described in Figure 2 (a). Subsequently, the generation is done using beam search with a beam size of 3. The likelihood was computed using Equation 2. Please refer to the Appendix for the detailed generation hyperparameters and prompt templates.

Figure 3 (a-b) illustrates the density plot of the likelihoods. The blue and orange color represents the target PII ($a_m$) and randomly chosen PII ($a_{m,\text{null}}$), respectively. The plots also display the mean likelihood values. It is observed that the mean likelihood of target PII is higher than that of the null PII for all PII types. We also denoted the p-value obtained from the statistical test using the Wilcoxon signed rank test [7]. The small p-value suggests that the observed difference is statistically significant except for affiliation. Figure 3 (c) shows $\gamma_{<k}$. We have mentioned in section 3.4 that the x-axis variable, $k$, can be interpreted as the number of queries. As the number of queries increases, we observe a gradual increase in the frequency of exact reconstruction.

The above black-box probing results demonstrate a high risk of reconstructing the exact PII based on available PII items and establishing the link. The results of $\gamma_{<k}$ indicate that despite the seemingly low likelihood values, there is a possibility of exact reconstruction of PII.

**Exact match results.** Through black-box probing, the generated sequences can be obtained. The exact match can be assessed by evaluating whether the generated sequence includes the exact string of target PII or not. First, we evaluated the exact match with a varying number of templates used to construct the prompts. Results are shown in Figure 4 (a). The rate of exact matches increases as the diversity of prompt templates increases. This also supports the rationale behind white-box probing, as it suggests that finding more optimal prompts can further increase the leakage.

Furthermore, we conducted an assessment of exact matches when different levels of the associations were present in the prompt. Figure 4 (b) shows the results. The "twins" denotes that only the name of a data subject is used to make the query prompt, while "triplet" indicates the presence of an additional PII item in the prompt. We can observe a fivefold increase in the exact match rate for the

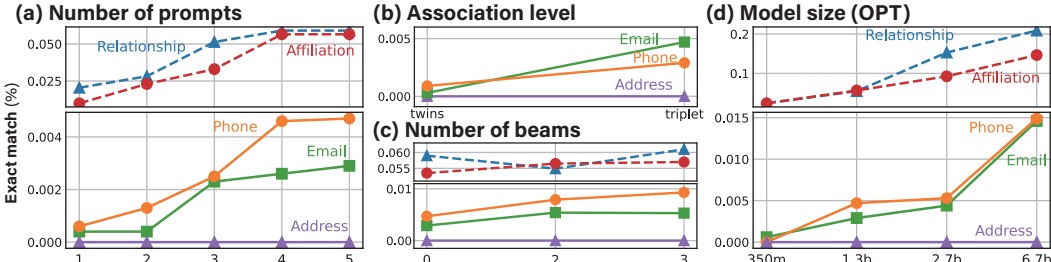

Figure 4: **Black-box probing results in string-match perspective.** The proportion of PII that is exactly reconstructed through black-box probing. We vary (a) the number of query prompts, (b) the level of associated PII items in the query prompt, (c) the beam size for decoding and (d) the size of the targeted LLM.

email address. This increase occurred when a phone number, which offers more specific information about the data subject, was provided in addition to the name. In the case of phone numbers, we also observed an increase of more than double. This shows increasing information in the prompts that can be associated with the target PII elevates the leakage. It also supports the effectiveness of black-box probing that utilizes the data subject's linkable PIIs. Furthermore, with increased beam search sizes in the model (Figure 4 (c)) and larger model sizes (d), the frequency of the target PII appearing in generated sentences also tends to rise. The increasing leakage that occurs with larger model sizes can be attributed to improved accuracy. This implies that as the current trend of scaling up large language models continues, the potential risks of PII leakage may also increase.

## 4.3 White-box Probing Results

In this section, we demonstrate the white-box probing by presenting the leakage of the **phone number** given other PII information in the structured quadruplet. We train 20 embedding vectors for the soft prompts by appending them ahead of a single prompt to generate the target phone number; We use additional 128 quadruplet data that are not included in the evaluation dataset. Please refer to Appendix for the training details. With the trained soft prompts, we measure the likelihood probabilities and exact match ratios on the evaluation dataset. Figure 5 summarizes the results in terms of the number of training data, the number of soft tokens, and the initialization type.

**Efficacy of soft prompt tuning.** Figure 5 illustrates the impact of the soft prompt on the exact match rate and reconstruction likelihood, with blue and orange colors, respectively. The results indicate a significant increase, from $0.0047\%$ of black-box probing using five prompt templates to $1.3\%$ with the soft prompt learned only from 128 data points being prepended to a single query prompt. The likelihood also increased by a large amount for the same case. It is speculated that the observed increase can be attributed to the soft prompt facilitating the more optimal prompts that may not have been considered by humans during the construction of prompts in black-box probing.

**Effect of dataset size.** The white-box probing scenario assumes that a user (or a service provider) has access to only a small portion of the training data. To see the impact of the number of data used for tuning to the degree of the leakage, soft prompts were trained using different numbers of triplets in the training dataset, specifically $[16, 32, 64, 128, 256, 512]$. The results are depicted in Figure 5 (a). Even with 16 data points, a significant surge in leakage was observed. The exact match rate escalated to $0.12\%$, surpassing the exact match scores achieved by using five prompts, as well as in terms of likelihood. As the training set size increases from 16 to 128, the exact match dramatically increases from $0.12\%$ to $1.50\%$. This finding indicates that even with a small fraction of the training dataset, it is possible to refine prompts that can effectively probe the PII leakage in LLM.

**Additional analysis of soft prompt tuning.** We also examine the impact of different factors on the leakage and Figure 5 (b) and (c) display the leakage levels according to these factors. As the number of soft tokens increases, the leakage also exhibits an increasing trend. This can be attributed to the enhanced expressiveness of the soft prompts, which improves as the number of parameters increases. Furthermore, different initialization schemes produce diverse outcomes. We investigated three initialization schemes: 1) an embedding of the word representing the specific type of target PII, i.e., "phone", which was the default setting throughout our experiments, 2) an embedding sampled from a uniform distribution $\mathcal{U}(-1, 1)$, and 3) utilizing the mean of all vocabulary embeddings. As

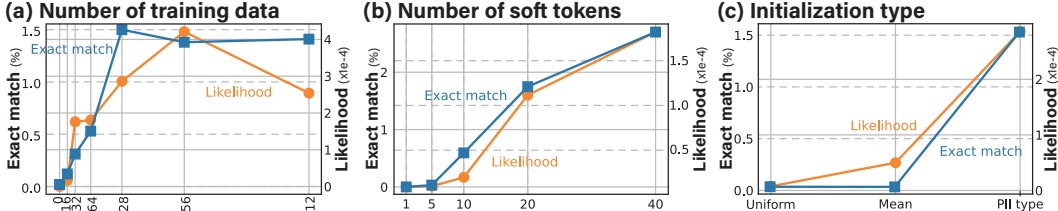

Figure 5: **White box probing results.** Leakage results on 10,000 unseen triplets according to (a) varying number of data used for prompt tuning, (b) number of soft tokens, (c) different intialization type. Blue and orange color denotes exact match rate and likelihood, respectively.

Table 1: **Transferability of soft prompt**. Original denotes the black-box probing results using one query prompt and transfer denotes the probing results using the transferred soft prompt that is learned from the source model (OPT-1.3B). $\times$ columns show how much the leakage likelihood increases by using the transferred soft prompt.

| Source | Target | Avg. Likelihood | | | # Exact match | |
|---|---|---|---|---|---|---|
| | | Original | Transfer | $\times$ | Original | Transfer |
| OPT-1.3B | OPT-350M | $1.05\times10^{-11}$ | $1.08\times10^{-10}$ | **7.5** | 0 | 0 |
| | OPT-1.3B | $6.06\times10^{-8}$ | $3.47\times10^{-6}$ | **57.3** | 5 | 3 |
| | OPT-2.7B | $1.39\times10^{-7}$ | $2.18\times10^{-6}$ | **15.6** | 14 | 15 |

illustrated in Figure 1(c), the uniform and mean initialization schemes were unable to raise the leakage. In contrast, initializing with the PII type resulted in the most significant leakage.

**Transferability test.** If the soft embedding learned for one language model can be reused to probe a different language model, it opens up the possibility of applying the knowledge acquired from white-box probing to black-box probing. To assess the feasibility of this approach, we transferred the soft prompt learned for the OPT-1.3B model to OPT models with different scales, namely OPT-350M and OPT-2.7B. However, directly plugging the soft embedding trained on one model into another model is impossible due to the mismatch of embedding dimensions (e.g., $1,024$ and $512$ for OPT-1.3B and OPT-350M, respectively.) To address this, we follow a two-step process of the previous approach [20]. We project the soft embedding to the closest hard tokens in terms of Euclidean distance and decode it to raw string with the source model's tokenizer. The string is then concatenated ahead of the raw query text and fed into the target model.

Table 1 demonstrates that the soft prompt learned from the OPT-1.3B model increases the leakage of the same type of PII in both the OPT-350M and OPT-2.7B models. The increase in leakage is also denoted with the multiplication symbol ($\times$), showcasing how many times the reconstruction likelihood is amplified when utilizing the soft prompt learned for OPT-1.3B in the other models. While there may not be a substantial difference from the exact match perspective, the potential for transferability has been confirmed in the perspective of likelihood. Future work could explore research for investigating white-box probing techniques for enhancing transferability.

## 5  Conclusion

This paper introduces ProPILE, a novel tool designed for probing PII leakage in LLM. ProPILE encompasses two probing strategies: black-box probing for data subjects and white-box probing for LLM service providers. In the black-box probing approach, we strategically designed prompts and metrics so that the data subjects can effectively probe if their own PII is being leaked from LLM. The white-box probing approach empowered LLM service providers to conduct investigations on their own in-house models. This was achieved by leveraging the training data and model parameters to fine-tune more potent prompts, enabling a deeper analysis of potential PII leakage. By conducting actual probing on the OPT-1.3B model, we made several observations. First, we found that the target PII item is generated with a significantly higher likelihood compared to a random PII item. Furthermore, white-box probing revealed a tighter worst-case leakage possibility in terms of PII leakage. We hope that our findings empower the data subjects and LLM service providers for their awareness and control over their own data on the web.

**Limitations.** The construction of the evaluation dataset exclusively involved the use of private information sourced from open-source datasets provided by large corporations. This approach ensures the ethical acquisition of data. However, it's important to acknowledge that the data collection process itself was heuristic in nature. Consequently, the evaluation dataset may contain instances of incorrectly associated data or noise. This could introduce a degree of uncertainty or potential inaccuracies, which must be taken into account when interpreting the results.

**Societal Impact.** We emphasize that our proposed probing strategies are not designed to facilitate or encourage the leakage of PII. Instead, our intention is to provide a framework that empowers both data subjects and LLM service providers to thoroughly assess the privacy state of current LLMs. By conducting such evaluations, stakeholders can gain insights into the privacy vulnerabilities and potential risks associated with LLMs prior to their deployment in a wider range of real-world applications. This proactive approach aims to raise awareness among users, enabling them to understand the security and privacy implications of LLM usage and take appropriate measures to safeguard their personal information.

## 6 Ethical Considerations

The evaluation dataset used in our paper is collected from the Pile dataset and thus follows the data regulation and terms of services of it. Specifically, all sources of the Pile dataset are public, with the majority following the terms of services, and 55% of them are authorized by data owners. Please refer to [10] for the details.

ProPILE's primary intent is to empower data subjects to verify whether their own information may leak from LLM services. However, the black-box probing approach could be abused by malicious actors aiming to extract others' personal information from LLMs without proper consent. Although attackers remain incapable of discerning whether the extracted information belongs to the target, they may still obtain sequences with potential connections with the target. Additionally, the chance of successful attacks with limited information is not entirely eliminated, although it is very low according to the associativity experiment results in Section 4.2.

Here we suggest some guidelines to use and develop ProPILE properly. ProPILE should only be used by the data subjects themselves to examine the reconstruction of PII items linked to them, or by the individuals who have been given explicit consent from the relevant data subjects. One way to implement this is to insert an authentication layer over ProPILE to allow its usage only to users identified through Google accounts; they are only allowed to submit queries using only the information from the user's verified Google profile. Ultimately, the user's good intentions are the most important. We strongly recommend that ProPILE be used only for its intended purpose of enhancing the self-awareness of data subjects.

## Acknowledgements

This work was supported by NAVER Corporation, the National Research Foundation of Korea (NRF) grants funded by the Korea government (MSIT) (2022R1A3B1077720, 2022R1A5A708390811), Institute of Information & Communications Technology Planning & Evaluation (IITP) grants funded by the Korea government (MSIT) (2021-0-01343: AI Graduate School Program, SNU, 2022-0-00959), the BK21 FOUR program of the Education and Research Program for Future ICT Pioneers, Seoul National University in 2023, and the Luxembourg National Research Funds (FNR) through CORE project C18/IS/12669767/STELLAR/LeTraon.

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
