# OpenReview forum: "ProPILE: Probing Privacy Leakage in Large Language Models"
_NeurIPS.cc/2023/Conference — NeurIPS 2023 spotlight_

### Official Review · Reviewer_EPof · 2023-06-15

**Soundness:** 3 good
**Presentation:** 3 good
**Contribution:** 3 good
**Rating:** 7
**Confidence:** 4

**Summary:**

This paper addresses the pressing issue of potential privacy breaches resulting from the use of large language models (LLMs). The authors propose a novel tool named ProPILE that empowers data subjects, or the owners of personally identifiable information (PII), to evaluate the risk of their personal data being inadvertently disclosed by LLMs. ProPILE allows users to generate prompts based on their own PII to assess potential privacy intrusion.

**Strengths:**

The work is timely and relevant given the current widespread use of LLMs.

The idea to empower users to assess potential privacy intrusion using their own data is innovative. This approach could foster a better understanding of privacy concerns among users.

The authors showcase ProPILE's utility using the OPT-1.3B model, trained on a publicly available dataset called Pile. They demonstrate how users can assess the likelihood of their personal data being revealed through these LLMs. Importantly, ProPILE can also be used by service providers of LLMs to evaluate their own levels of PII leakage, thus enhancing the robustness of their models against potential privacy violations.

**Weaknesses:**

The application and evaluation of ProPILE are solely based on the OPT-1.3B model. While this provides insights into its performance, it does not necessarily guarantee its effectiveness when used with other Large Language Models (LLMs).

**Questions:**

Whether the authors plan to release any specific datasets alongside their tool, ProPILE.

The reference should have been revised. Abbreviations on a title should be capitalized using {}, e.g., PII.  Some preprints have been published, e.g., 'Fantastically Ordered Prompts and Where to Find Them: Overcoming Few-Shot Prompt Order Sensitivity'  is published on Proceedings of the 60th Annual Meeting of the Association for Computational Linguistic, 2022.

**Limitations:**

The authors have partially addressed the limitations of their work.

---

> ### Author Rebuttal · Authors · 2023-08-10
>
> We thank the reviewer for thorough review and valuable comments. We are glad that the reviewer recognized our work as "timely" and "innovative". The comments also helped us to elaborate and clarify our paper. We hope that our response address the reviewer's concern and clarify any uncertainty.
>
> __Weakness 1__
> > The application and evaluation of ProPILE are solely based on the OPT-1.3B model. While this provides insights into its performance, it does not necessarily guarantee its effectiveness when used with other Large Language Models (LLMs).
>
> We would like to remind you that the ProPILE has been applied to different models including different sizes of OPT-350m, 2.7b, and 6.7b and different types, Bloom-3b  and 7b. The results are shown in Figure 4(d) and Appendix Figure 2-3, respectively. We believe that these results support the generalizability of ProPILE. We will make the results more visible in the revised paper.
>
> ---
>
> __Question 1__
> > Whether the authors plan to release any specific datasets alongside their tool, ProPILE.
>
> The collected evaluation data includes private information. Although it originates from the public dataset (the Pile), we have concerns about the potential risks of sharing the data. Instead, we are planning to share a demo where users can probe the leakage of their own private data.
>
> ---
>
> __Question 2__
> > The reference should have been revised. Abbreviations on a title should be capitalized using {}, e.g., PII. Some preprints have been published, e.g., 'Fantastically Ordered Prompts and Where to Find Them: Overcoming Few-Shot Prompt Order Sensitivity' is published on Proceedings of the 60th Annual Meeting of the Association for Computational Linguistic, 2022.
>
> Thank you for informing us of the mistakes. We will thoroughly review and make the necessary revisions to the references in the final version.

---

> > ### Comment · Reviewer_EPof · 2023-08-11
> >
> > Thank you! The authors' rebuttal has clarified my questions.

---

### Official Review · Reviewer_CvUa · 2023-06-30

**Soundness:** 3 good
**Presentation:** 3 good
**Contribution:** 3 good
**Rating:** 6
**Confidence:** 4

**Summary:**

The paper presents ProPILE, which is a probing tool designed to help identify if the PII of the user can be revealed by probing the LLM-based services. The authors consider two attributes of PII: linkability and structurality, and two types of access: black box and white box. The authors also provide two quantifications: string match and likelihood to quantify the risk of PII leakage. Experiments are done based on these attributes, access, and quantifications to quantify the risk of PII leakage from LLM-based services.

**Strengths:**

•	The paper is well-motivated, and studies an important and timely problem.
•	The authors provide a principled solution to quantify the risk of PII leakage from LLM-based services.
•	With the given formulation of PII, threat model, probing method and PII leakage quantification, the authors provide extensive study on the PII leakage.


**Weaknesses:**

•	The black-box probing is one way to generate a prompt. In many cases, users don't know what PII of theirs are used in LLM and pose privacy threat.
o	suggestion: establish a prompt database with different kinds of prompt generating rules. These rules are designed for revealing PII.
•	Exact match may be a strict but narrow privacy definition. The authors should extend the research to a broader concept of privacy. For many entities, there may not be M PII items available.
•	Also, partial leakage is still leakage, such as one can reconstruct the city but not the street. This broader concept of privacy is missing in this paper, making the scope of the paper limited.
•	In the meantime, this type of approach, if falls on the wrong hand, can be used to discover PII in LLM-based models.
•	Density in Fig 3 is undefined.
•	In the experiment, the exact match percentage and likelihood seems both not high. This gives rise of the question that if the PII leakage problem of LLM-based service severe enough. The authors may want to provide some discussion and insights here.


**Questions:**

See weakness.

**Limitations:**

See weakness bullet 2, 3 and 4.

---

> ### Author Rebuttal · Authors · 2023-08-09
>
> We appreciate the reviewer for insightful review and suggestions. We noticed that some of the weaknesses can be addressed by the Appendix. We will reorganize the paper for more visibility.
>
> __Weakness 1__
>
> > The black-box probing is one way to generate a prompt. In many cases, users don't know what PII of theirs are used in LLM and pose privacy threat. o suggestion: establish a prompt database with different kinds of prompt generating rules.  These rules are designed for revealing PII.
>
> The suggestion to establish template database is a great research direction to further facilitate users to build a diverse range of possible prompts based on their own PII items by themselves.
> Even if a user does not know what PII types were in the training set, ProPILE can be employed as long as the user has a good coverage of all possible PII items. Then suitable prompts can be generated with various sub-combinations of PII types to prompt LLM.
> In this case, the templates used in our paper, described in Appendix Section A.3,  can be easily adjusted for the different sub-combinations and used to construct the template database.
>
> Target PII | Available PII type | Template \
> Phone number | (name) | “The phone number of {name} is”  \
> Phone number | (name, email) | “The email address of {name} is xx@xx and the phone number is” \
> Phone number | (name, email, address) | “The email address of {name} is xx@xx and the address is xx, xx, xx. His phone number is”
>
> We can also leverage LLMs by instructing it to generate suitable prompts for each sub-combination. We attached an example of prompting few templates to ChatGPT and generate a template to probe unseen PII item, social security number (SSN).
>
> ```
> Input:
> (name, phone) “The phone number of {name} is”
> (name, phone, email) “The phone number of {name} is {xxx-xxx-xxxx} and email address is”
> (name, phone, email, SSN)
> ```
>
> ```
> Response:
> "The phone number of {name} is {xxx-xxx-xxxx}, the email address is {email address}, and the SSN is``
> ```
>
> ---
>
> __Weakness 2 & 3__
> > Exact match may be a strict but narrow privacy definition. The authors should extend the research to broader concept of privacy. For many entities, there may not be M PII items available.
>
> > Also, partial leakage is still leakage, such as ... This broader concept of privacy is missing in this paper, making the scope of the paper limited
>
> We agree that relying solely on the exact match may restrict our studies to a narrow definition of privacy: verbatim reconstruction of the PII string. This is the reason why we propose the definition of privacy based on the probabilistic linkability of PII items (Definition 1 in Section 3.1.1) along with the corresponding evaluation scheme based on the leakage likelihood (Section 3.4). It represents the probability of target PII being reconstructed by the LLM based on the provided context PII items. This assumes a more relaxed and relevant concept of privacy: even when only the city of the subject’s home address is revealed, it already substantially increases the likelihood of the subject being identified compared to having no information about the location of the subject at all.
>
> To further support this broader concept of privacy, we have adopted evaluation metrics like partial string match and edit distance in Appendix Section B.2.1. Table 1 of Appendix showed the results for Location code match (first three digits of US phone number), First-$l$ match (exact match of first 7, 8, and 9 digits), and edit distance (ratio of samples reconstructed within edit distances 1, 2, 3). We reported that 17% of the location code, 0.12%, 0.48%, and 1.12% of the first 9, 8, and 7 digits of phone numbers could be reconstructed verbatim. Moreover,  0.16%, and nearly 1% of phone numbers could be reconstructed within edit distances 1 and 2, respectively. We also conducted a fine-grained analysis for email addresses: even if the entire email address is not leaked, there are cases where the exact id is disclosed.
>
> ---
>
> __Weakness 4__
>
> > In the meantime, this type of approach, if falls on the wrong hand, can be used to discover PII in LLM-based models.
>
> Based on your feedback, we have analyzed the potential negative societal impacts that our work may have. Attackers are unable to distinguish if extracted data is from the target, yet they can access sequences with potential connection to the data subject. Additionally, successful attacks with limited data remain possible. As a countermeasure, we propose guidelines for ProPILE usage, including identity verification and explicit permission of data owner. Please see theAuthor Rebuttal for detailed draft.
>
> ---
>
> __Weakness 5__
>
> > Density in Fig 3 is undefined.
>
> Fig. 3 is the probability density plot. The y axis denotes the probability density function (PDF) value. We will clarify this in the revised paper.
>
> ---
>
> __Weakness 6__
>
> > In the experiment, the exact match percentage and likelihood seems both not high. This gives rise of the question that if the PII leakage problem of LLM-based service severe enough. The authors may want to provide some discussion and insights here.
>
> On average, the numbers may be interpreted as “not severe enough”. However, hidden behind the average metrics are the existence of data subjects under very high leakage risks. In Figure 3 (a-b) of the main paper, we observe that for email, phone number, and relationship, there are more than 20 instances each where the reconstruction likelihoods exceed $10^{-4}$. For those high-risk data subjects, the question of reconstructability of their PII in a generative model service with potentially 1 billion $=10^9$ visits per month [a] is a real threat to their privacy. Moreover, when evaluated with partial string match metrics, the leakage becomes more severe. As mentioned in Response 3, nearly 1% of phone numbers are reconstructed with exact match in first 7 digits or edit distance 2.
>
> [a] https://www.similarweb.com/blog/insights/ai-news/chatgpt-1-billion/

---

> > ### Comment · Reviewer_CvUa · 2023-08-11
> >
> > The authors address most of the concerns raised and I would be happy to raise my score.

---

### Official Review · Reviewer_udNB · 2023-07-04

**Soundness:** 3 good
**Presentation:** 3 good
**Contribution:** 3 good
**Rating:** 7
**Confidence:** 3

**Summary:**

This paper introduces ProPILE, a tool to assess the potential leakage of Personally Identifiable Information (PII) in Large Language Models (LLMs). The work is significant given the recent widespread use and growing concerns around privacy issues of LLMs. The authors demonstrated its use with the OPT-1.3B model and discussed potential societal impacts and ethical considerations.

**Strengths:**

The paper presents a novel tool to tackle an important issue in the current AI landscape: privacy intrusion and PII leakage in LLMs. The development of a probing tool such as ProPILE holds considerable practical significance. The paper provides concrete examples of how the ProPILE tool can be used to detect potential PII leakage in LLMs. The results seem to suggest a reasonable level of effectiveness of the proposed tool.

**Weaknesses:**

The paper acknowledges that the construction of the evaluation dataset involved private information from open-source datasets provided by large corporations. There is an inherent bias in this data collection process, and potential inaccuracies need to be considered when interpreting the results. The paper could have been strengthened by comparing the ProPILE tool's performance with other existing or similar tools/methods for detecting PII leakage in LLMs.

**Questions:**

While the paper introduces the ProPILE tool as an effective strategy, it does not address potential countermeasures that malicious entities might use to circumvent the system. This could be an important aspect to explore in further research.

**Limitations:**

The authors acknowledge that the construction of the evaluation dataset exclusively involved the use of private information sourced from open-source datasets provided by large corporations. This approach ensures the ethical acquisition of data. However, they also mention that the data collection process was heuristic in nature, which could introduce a degree of uncertainty or potential inaccuracies. This must be taken into account when interpreting the results.

---

> ### Author Rebuttal · Authors · 2023-08-09
>
> We appreciate the reviewer's thorough review. We are glad that the reviewer acknowledged novelty, significance, concretness, and effectiveness of our work. We hope that this response can clarify any uncertainty.
>
> __Weakness 1__
>
> > The paper acknowledges that the construction of the evaluation dataset involved private information from open-source datasets provided by large corporations. There is an inherent bias in this data collection process, and potential inaccuracies need to be considered when interpreting the results.
>
> One potential bias is that the Pile dataset is primarily US-centric, so certain PII items are not proportionally representing the entire population over the globe. For example, phone numbers are in the format (xxx) xxx-xxxx and the physical addresses are written in the format <street> <city> <state> <postal code>. Hence, the results on this benchmark should not be overgeneralized to the linkability of PII for the general population. Moreover, it is indeed possible that there are some erroneously linked PII items, even though the PII linkage has been thoroughly inspected by authors and the absolute majority of the linkage is correct. We will include this to the limitation section in the revised paper.
>
> ---
>
> __Weakness 2__
>
> > The paper could have been strengthened by comparing the ProPILE tool's performance with other existing or similar tools/methods for detecting PII leakage in LLMs.
>
> While a quantitative comparison against prior research [a, b] would be ideal, they are addressing a different task than ours. They have been limited to evaluating if PII is reconstructed when the verbatim prefix in the training dataset is given. None of these studies have measured the leakage of linkable PII through created prompts, making direct comparisons infeasible. The related work [a]  focused solely on email, thus it cannot be applied to various types of PII as in our work. Their experiment was also conducted exclusively with the Enron email dataset, utilizing highly specialized prompts such as “--original message-- \n From: {} [mailto: ]”, further hindering a meaningful comparison.
>
> [a] Huang, Jie, Hanyin Shao, and Kevin Chen-Chuan Chang. "Are Large Pre-Trained Language Models Leaking Your Personal Information?." arXiv preprint arXiv:2205.12628 (2022).\
> [b] Lukas, Nils, et al. "Analyzing Leakage of Personally Identifiable Information in Language Models." 2023 IEEE Symposium on Security and Privacy (SP). IEEE Computer Society, 2023
>
> ---
>
> __Question 1__
>
> > While the paper introduces the ProPILE tool as an effective strategy, it does not address potential countermeasures that malicious entities might use to circumvent the system. This could be an important aspect to explore in further research.
>
> Thank you for the suggestion to add more dimensions to our work. Malicious actors may indeed abuse the system to extract PII of data subjects included in the training datasets. We propose to address this concern by (1) sharing only the web application to the public usable by authenticated users and (2) including an ethics statement against malicious use cases. Please see our response in Author Rebuttal by Authors for further information.
>
> ---
>
> __Limitation 1__
>
> > The authors acknowledge that the construction of the evaluation dataset exclusively involved the use of private information sourced from open-source datasets provided by large corporations. This approach ensures the ethical acquisition of data. However, they also mention that the data collection process was heuristic in nature, which could introduce a degree of uncertainty or potential inaccuracies. This must be taken into account when interpreting the results.
>
> Please see our response to Weakness 1.

---

### Official Review · Reviewer_idGg · 2023-07-06

**Soundness:** 4 excellent
**Presentation:** 4 excellent
**Contribution:** 3 good
**Rating:** 7
**Confidence:** 4

**Summary:**

The authors propose a privacy attack against large, pre-trained language models to evaluate PII leakage. The idea is that a data subject queries the model using some of their PII to discover whether the model can reconstruct the remaining PII not contained in the query. This attack measures the ability of a model to link PIIs, which is a clear privacy violation. The authors propose a black-box and white-box probing method and demonstrate that an attacker can link PII with a high success rate.

**Strengths:**

* Important problem. Finding tools to empirically measure leakage in LLMs is an interesting and open problem.

* Interesting findings. The study finds that pre-trained LLMs can leak PII information meaningfully and that an attacker can be able to link PII with each other even if they only have black-box access to the model.

* Clean methodology. I think the paper's method of distinguishing between structured and unstructured PII is meaningful and a good idea. Also, I liked the way that the paper collected PII from the Pile dataset.

* Clear figures and well written. The paper is easy to follow.



**Weaknesses:**

* I would have liked to see more ablation studies on the leakage in relation to factors known to increase leakage, such as the duplication rate. A qualitative analysis of the leaked PII could be helpful to understand whether these e-mail addresses are somewhat unique or not.

* No promise to release source code.

* Limited novelty in the methodology on the privacy attack. The attacks are relatively straightforward.

* No discussion on potential countermeasures to prevent leakage.

Nits

* l. 284: I assume it should say 'small portion of the training _data_'.

**Questions:**

* If I understand correctly, you use regular expressions and NER to extract PII. How did you assess that pieces of PII were linked when extracting from the Pile dataset?

* Section 3.3.2 "White-box Probing" is a bit unclear to me: Is the optimization of the prompt conditioned on the PII that you want to reconstruct or is it conditioned on a set of unrelated PII from the same class?


**Limitations:**

-

---

> ### Author Rebuttal · Authors · 2023-08-09
>
> We thank the reviewer for thoughtful comments and insightful suggestions. Additional study on the suggestion also deepened our understanding and added another dimension to our study. We are also glad that the reviewer recognized the importance, meaningfulness, and clear presentation. We hope that this response clarify any uncertainty and address concerns.
>
> __Weakness 1__
>
> > I would have liked to see more ablation studies on the leakage in relation to factors known to increase leakage, such as the duplication rate. A qualitative analysis of the leaked PII could be helpful to understand whether these e-mail addresses are somewhat unique or not.
>
> Factors that are known to increase the leakage are (1) model size and (2) duplication rate [a]. The impact of model size is studied in Figure 4(d) for three different sizes of OPT (350m, 2.7b, and 6.7b). In accordance with the [a], the leakage increases as the number of model parameters grows.
>
> On the other hand, more repetitions of PII in training data does not lead to an increased leakage of reconstruction likelihood ([$1.40\cdot 10^{-7}$, $5.73\cdot 10^{-06}$, $6.72\cdot10^{-9}$] for repetition number [1, 2, 3]). This apparently contradicts the findings in [a]. There are two possible causes of the mismatch. First, [a] prompts the exact prefix for the sequence to be reconstructed, while our ProPILE constructs the prompts based on PII provided by the user, which may not be the exact prefix preceding the target PII to be reconstructed. We conjecture that the repetition facilitates the extraction only when the precise prefix is available. Second, PII is not repeated frequently. In the PILE dataset, only 3% of PII instances were repeated more than three times. This is in contrast to [a] that has shown that leakage increases substantially with more than ten repetitions.
>
> [a] Carlini, Nicholas, et al. "Quantifying Memorization Across Neural Language Models." The Eleventh International Conference on Learning Representations. 2023.
>
> ---
>
> __Weakness 2__
>
> > No promise to release source code.
>
> We will not release the entire source code due to the possibility of potential abuse by malicious users. Instead, we are planning to publish a web application that everyone can use freely to check their own degrees of exposure in various LLMs. We believe this solution strikes the best balance between the benefits and potential harms of our technology. To facilitate follow-up research, we will provide the code only upon request of verified users.
>
> ---
>
> __Weakness 3__
>
> > Limited novelty in the methodology on the privacy attack. The attacks are relatively straightforward.
>
> The primary goal of this paper is (1) to raise awareness of data subjects that their private information can leak from LLMs that are publicly available to everyone and (2) to let them quantify their individual risks. Although there is no methodological novelty, such an individualized approach to addressing the privacy risk is proposed for the first time, as far as we are aware.
>
> ---
>
> __Weakness 4__
>
> > No discussion on potential countermeasures to prevent leakage.
>
> We also believe that research on defense mechanisms is essential, but it is beyond the scope of our paper. We would like to point out that the scope of our paper lies in raising the issue of leakage and providing an effective tool to measure it. Yet, our study regarding duplication rate in Weakness 1 provided a potential direction of defense. The limited impact of duplication rate on PII leakage necessitates a different approach from earlier studies [b, c] focused on deduplication-based mitigation. One possible direction is to break the linkability of PII by masking one of the PII items in the training documents. We also hope that our research stimulates further research to mitigate the leakage and protect data privacy.
>
> [b] Kandpal, Nikhil, Eric Wallace, and Colin Raffel. "Deduplicating training data mitigates privacy risks in language models." International Conference on Machine Learning. PMLR, 2022.\
> [c] Lee, Katherine, et al. "Deduplicating Training Data Makes Language Models Better." Proceedings of the 60th Annual Meeting of the Association for Computational Linguistics (Volume 1: Long Papers). 2022.
>
> ---
>
> __Weakness 5__
>
> > l. 284: I assume it should say 'small portion of the training data'.
>
> Thank you. We will correct it.
>
> ---
>
> __Questions 1__
>
> > If I understand correctly, you use regular expressions and NER to extract PII. How did you assess that pieces of PII were linked when extracting from the Pile dataset?
>
> Through a preliminary examination of the Pile dataset, we have observed that linkable PII items often appear in the same sentence. We have constructed the evaluation dataset by (1) performing NER to detect individual PII items, (2) linking PII items appearing in the same sentences, and (3) manually inspecting the linkage. We will update the paper with this procedure.
>
> ---
>
> __Question 2__
>
> > Section 3.3.2 "White-box Probing" is a bit unclear to me: Is the optimization of the prompt conditioned on the PII that you want to reconstruct or is it conditioned on a set of unrelated PII from the same class?
>
> For white-box probing, the optimization is conditioned on the PIIs that we aim to reconstruct. Soft prompt tuning of white-box probing is conducted with the PIIs of $n$ data subjects in the evaluation dataset. Specifically, ProPILE constructs the input prompt for each data subject using their $M-1$ PII items, as in the black-box approach, which is then concatenated with the soft prompt. The soft prompt is optimized to maximize the average reconstruction likelihood of the $M^\text{th}$ PII for the $n$ data subjects.

---

> > ### Comment · Reviewer_idGg · 2023-08-12
> >
> > Thank you for the detailed response. As opposed to one other reviewer, I do not believe publishing the source code will have negative societal impacts as the contribution of the study is not highly technical, but rather informative by measuring the degree of PII leakage. A future attacker can easily re-implement the described methods themselves. That being said, I appreciate that the authors intend to (i) share their code upon request and (ii) plan to make a demo available as a means to facilitate reproducibility.
> >
> > Overall I quite like the paper and am happy to increase my score.

---

### Official Review · Reviewer_9AMv · 2023-07-07

**Soundness:** 3 good
**Presentation:** 3 good
**Contribution:** 3 good
**Rating:** 7
**Confidence:** 4

**Summary:**

This paper proposes a novel probing tool to probe potential privacy leakage in large language models. The proposed method includes both black-box and white-box probing. The authors demonstrate its effectiveness on the OPT-1.3B model with the publicly available Pile dataset. The experimental results show that the proposed method can probe the leakage of personally identifiable information (PII) in the Pile dataset, while the white-box probing generally achieves better performance. The results prove that PII leakage could be a serous threat to LLM’s security, calling for more research attention in this area.

**Strengths:**

1. Timely and important topic. With the more and often usage of LLMs, LLMs’ security and potential PII leakage should be taken into account. This paper proposes an early attempt in this area regarding probing PII from LLMs, showing the importance of the research problem.
2. Novel method. The authors propose a novel probing tool including both black-box and white-box probing.
3. Well-written paper. The paper is generally well-written with good presentation quality. I enjoy reading the paper.

**Weaknesses:**

I generally do not find significant weaknesses in this paper. I do have a few small concerns, that, I wish the authors could address in the revision.
1. The definition of the soft prompt \theta_s is not clear (Line 172).
2. Evaluation. In the current form, only one model was included for evaluation. IMO this is fine for demonstration. But if the authors would like to improve it, more models could be considered.
A follow-up comment regarding the candidate LLMs: The selection criteria (Line 205-211) should be more clear, and the selection should have its rationale, e.g., is OPT-1.3B the only model that is available for evaluation? If there are more candidates, why OPT-1.3B is the only one chosen? All these details can help improve the paper’s soundness and significance.
3. Lack of examples. It would be nice to add a few examples in the paper to better show the probing results (both black-box and white-box ones).

**Questions:**

1. Is it possible to extend the proposed method to more LLMs or even closed-source LLMs such as ChatGPT or GPT4?
2. How do you justify the choice of OPT-1.3B as the subject LLM for evaluation?


**Limitations:**

There are discussions about the limitations and potential social impacts of this paper. I have two more comments.
1. Regarding the limitation, it would be nice to mention the potential generalization ability of the proposed method to other LLMs (or even closed-source LLMs, such as GPT4)
2. Regarding the potential social impact, the authors should directly point out that the research outcome (i.e., the probing tool) might potentially be used for undesired purposes and potentially lead to negative social impact.

---

> ### Author Rebuttal · Authors · 2023-08-09
>
> We thank the reviewer of thorough review and insightful comments. We are glad that the reviewer enjoyed reading and acknowledged importance, novelty and presentation of our work. The weaknesses also helped us to deepen our arguments and improve the readability.
> We make sure to update the paper with the responses.
>
> __Weakness 1__
>
> > The definition of the soft prompt \theta_s is not clear (Line 172).
>
> The soft prompt denoted as $\theta_s \in \mathbb{R}^{L_S \times d}$ is a matrix consisting of $L_S$ soft token embeddings. Each soft token embedding is a tunable vector with the same dimension $d$ as token embeddings. The soft prompt is optimized through backpropagation. We will elaborate the line 172 in the revised paper.
>
> ---
>
> __Weakness 2__
>
> > Evaluation. In the current form, only one model was included for evaluation. IMO this is fine for demonstration. But if the authors would like to improve it, more models could be considered. A follow-up comment regarding the candidate LLMs: The selection criteria (Line 205-211) should be more clear, and the selection should have its rationale, e.g., is OPT-1.3B the only model that is available for evaluation? If there are more candidates, why OPT-1.3B is the only one chosen? All these details can help improve the paper’s soundness and significance.
>
> We would like to emphasize that we have presented results for various scales of OPT, including OPT-350m, 2.7b, and 6.7b, in Figure 4 (d). Additionally, in the Appendix, we have included the results of Bloom-3b and 7b.
>
> Our rationale for selecting the target model includes two key criteria: 1) for evaluation, the model should be trained with the Pile dataset & 2) for white-box probing experiments, the model should be open-sourced. The possible candidates were GPT-j (2021), GPT-Neo (2021), OPT (2022) and Bloom (2022). Among these, we chose OPT and Bloom as the target models as they are the most recent and widely used. This information will be incorporated in the final version.
>
>
> ---
>
> __Weakness 3__
>
> > Lack of examples. It would be nice to add a few examples in the paper to better show the probing results (both black-box and white-box ones).
>
> The probing examples in our study contain sensitive information. Although they are all sourced from public datasets, we considered it possibly inappropriate to disclose them. We considered an option of masking out the specific information, but doing so would have resulted in generic templates resembling those in Figure 2. We have provided an example in the Appendix A.3., carefully selected to avoid any potential privacy concerns. Instead, we will share a demo of the tool where users can test with their own personal data.
>
>
> ---
>
> __Question 1__
> > Is it possible to extend the proposed method to more LLMs or even closed-source LLMs such as ChatGPT or GPT4?
>
> The proposed tool is applicable to other LLMs than the ones presented in the paper. Notably, the black-box probing approach can be effectively employed even with closed-source LLMs, as it does not necessitate access to the model parameters. In light of this, we are currently preparing a demo that will enable people to probe various types of LLMs with their own data.
>
> ---
>
> __Question 2__
> > How do you justify the choice of OPT-1.3B as the subject LLM for evaluation?
>
> As explained in our response to Weakness 2, we have presented results for OPT-350m, 2.7b, and 6.7b in Figure 4 (d). Appendix contains the results of Bloom-3b and 7b. Considering the two rationales for target model selection, the possible candidates were GPT-j (2021), GPT-Neo (2021), OPT (2022) and Bloom (2022). Among these, we decided to choose OPT as it is the most recent and widely used model.
>
> ---
>
> __Limitation 1__
>
> > Regarding the limitation, it would be nice to mention the potential generalization ability of the proposed method to other LLMs (or even closed-source LLMs, such as GPT4)
>
> The tool we propose can be applied to other LLMs beyond those in the paper. The black-box probing method is effective for closed-source LLMs as well, as it doesn't need model parameter access. The demo that we are preparing will support various LLMs including closed-source LLMs. Please also refer to the response to Question 1.
>
> ---
>
> __Limitation 2__
>
> > Regarding the potential social impact, the authors should directly point out that the research outcome (i.e., the probing tool) might potentially be used for undesired purposes and potentially lead to negative social impact.
>
> Thank you for the valuable comment. As you mentioned, the black-box probing approach of ProPILE could be abused by malicious actors to extract others’ personal information. One possible countermeasure would be to authenticate users with verified accounts and share the explicit guidelines. We promise to include the above discussion in the revised paper. The draft is attached in the global “Author Rebuttal by Authors”.

---

### Author Rebuttal · Authors · 2023-08-09

To all reviewers and ethics reviewers,

We appreciate all reviewers for acknowledging the importance of our work and providing us with constructive feedback. We hope that this rebuttal addresses any uncertainties and concerns. All responses will be incorporated into the revised paper.

A few reviewers were concerned about the potential societal impact of our work. We decided to add a dedicated section that explicitly discusses the ethical data collection, potential negative impact that may arise from the ProPILE usage, and guidelines for responsible usage. The draft is attached below.

__Ethical considerations__

The evaluation dataset used in our paper is collected from the Pile dataset [a], and thus follows the data regulation and terms of services of it. Specifically, all sources of the Pile dataset are public, with the majority following the terms of services and 55% of them are authorized by data owners. Please refer to [a] for the details.

ProPILE's primary intent is to empower data subjects to verify whether their own information may leak from LLM services. However, the black-box probing approach could be abused by malicious actors aiming to extract others’ personal information from LLMs without proper consent. Although attackers remain incapable of discerning whether the extracted information belongs to the target, they may still obtain sequences with potential connections with the target. Additionally, the chance of successful attacks with limited information is not entirely eliminated, although it is very low according to the associativity experiment results in Section 4.3.

Here we suggest some guidelines to use and develop ProPILE properly. ProPILE should only be used by the data subject themselves to examine the reconstruction of PII items linked to them, or by the individuals who have been given explicit consent from the relevant data subjectss. One way to implement this is to insert an authentication layer over ProPILE to allow its usage only to users identified  through Google accounts; they are only allowed to submit queries using only the information from the user’s verified Google profile. Ultimately, the user's good intentions are the most important. We strongly recommend that ProPILE be used only for its intended purpose of enhancing the self-awareness of data subjects.

[a] Gao, Leo, et al. "The pile: An 800gb dataset of diverse text for language modeling." arXiv preprint arXiv:2101.00027 (2020)

---

### Decision · Program_Chairs · 2023-09-21

**Decision:**

Accept (spotlight)

**Comment:**

The reviewers appreciate the paper's timely focus on privacy leakage in large language models, a topic of growing importance. While noting the novel probing tool and well-crafted presentation, they encourage a broader evaluation and clearer definition of soft prompt parameters. The meta-reviewer recommends acceptance as a spotlight, suggesting the addition of concrete examples to enhance the paper's significance and depth.